# *N*-Heterocyclic Carbene Adducts of Alkynyl Functionalized 1,3,2-Dithioborolanes

**DOI:** 10.3390/molecules24091690

**Published:** 2019-04-30

**Authors:** Richard Böser, Lars Denker, René Frank

**Affiliations:** Department of Inorganic and Analytical Chemistry, Faculty of Life Sciences, Technische Universität Braunschweig, Hagenring 30, 38106 Braunschweig, Germany; R.Boeser@tu-braunschweig.de (R.B.); L.Denker@tu-braunschweig.de (L.D.)

**Keywords:** 1,3,2-dithioborolane, alkynyl boronate, *N*-heterocyclic carbene

## Abstract

Alkynyl functionalized boron compounds are versatile intermediates in the areas of medicinal chemistry, materials science, and optical materials. In particular, alkynyl boronate esters [R^1^−C≡C−B(OR^2^)_2_] are of interest since they provide reactivity at both the alkyne entity, with retention of the B−C bond or alkyne transfer to electrophilic substrates with scission of the latter. The boron atom is commonly well stabilized due to (i) the extraordinary strength of two B−O bonds, and (ii) the chelate effect exerted by a bifunctional alcohol. We reasoned that the replacement of a B−O for a B−S bond would lead to higher reactivity and post-functionalization in the resulting alkynyl boronate thioesters [R^1^−C≡C−B(S_2_X)]. Access to this poorly investigated class of compounds starts form chloro dithioborolane *cyclo*-Cl−B(S_2_C_2_H_4_) as a representative example. Whereas syntheses of three coordinate alkynyl boronate thioesters [R^1^−C≡C−B(S_2_X)] proved to be ineffective, the reactions of NHC-adducts (NHC = *N*-heterocyclic carbene) of *cyclo*-Cl-B(S_2_C_2_H_4_) afforded the alkyne substituted thioboronate esters in good yield. The products NHC−B(S_2_C_2_H_4_)(C≡C-R^1^) are remarkably stable towards water and air, which suggests their use as boron-based building blocks for applications akin to oxygen-based boronate esters.

## 1. Introduction

Alkynyl boronates [R^1^−C≡C−B(OR^2^)_2_] and alkynyl borates [R^1^−C≡C−B(OR^2^)_3_]^−^ (R^1^, R^2^ = hydrocarbyl moieties) can be viewed as ester of the parent alkynyl boronic acids R^1^-C≡C-B(OH)_2_ and are versatile building blocks, which are widely employed as intermediates for various applications in the areas of medicinal chemistry, materials science, and optical materials [1,2,3]. The key reactivity of these compounds includes (i) transformations at the alkyne entity retaining the B−C bond to afford borylated target compounds, or ii) the transfer of the alkyne moiety to electrophilic reagents with scission of the B−C bond [4,5,6,7,8,9]. Alkynyl boronates can be prepared from monovalent alcohols [10,11], but due to their increased hydrolytic stability the incorporation of bifunctional chelating alcohols, in particular 1,2-dihydroxy compounds, is highly preferred. Thus, the bivalent alcohols introduce a bridging backbone leading to improved stability, and the scope of 1,2-diols employed for alkynyl boronate esters includes pinacol (**1**) [12,13,14,15,16,17,18,19], catechol, (**2**) [20,21,22], and ethylene glycol (**3**) [23,24], Scheme 1. These compounds **1**–**3** may also be considered as boron substituted heterocycles and can be viewed as derivatives of the parent 1,3,2-dioxaborolane. Whereas oxygen is well established as a flanking neighbor atom at boron centers, the heavier chalcogen sulfur is less common. As indicated by the trend of bonds dissociation energies (BDE) for B−S (577 kJ/mol) vs. B−O bonds (810 kJ/mol) [25],the respective alkynyl boronate thioester (or 1,3,2-dithiaborolane) derivatives are expected to be more reactive in terms of B−S bond scission, which offers options for post-functionalization at the boron center [26]. In contrast to 1,3,2-dioxaborolane derivatives **1**–**3**, the analogous sulfur derivatives have been described in only one previous report in the form of 1,3,2-benzodithiaborole **4** containing an aromatic rigid scaffold [27]. Therefore, we were interested in broadening the scope of alkynyl 1,3,2-dithiaborolanes to an aliphatic backbone as a consistent extension of the oxygen-based 1,3,2-dioxaborolane formulate compounds of type **5** as target structures, Scheme 2.

## 2. Results

Since the series of compounds **4** could be prepared in acceptable yields (50%–65%) from halogenated 1,3,2-benzodithiaborole, we attempted to employ the established synthetic methodology for our target molecules **5** [27]. In analogous fashion, we reacted chloro dithiaborolane **6** [28] with a wide selection of silylated and stannylated alkynes. In contrast to the synthesis of compounds **4**, the ^11^B-NMR spectra (NMR = nuclear magnetic resonance)of the crude product only showed various broad signals in the range of 10–70 ppm, and the isolation of single compounds by sublimation or crystallization proved ineffective. Similar results were obtained when **6** was reacted with alkynyl organometallic reagents of more electropositive metals. Reactions performed with the alkynyl lithium reagents R−C≡C−Li (R = SiMe_3_ or Ph) in toluene (at ambient temperature or −78 °C) gave inseparable mixtures, while Grignard reagents of type R−C≡C−MgX (R = SiMe_3_ or Ph, X = Cl) in etherical solvents [Et_2_O, THF (tetrahydrofuran) —at ambient temperature or −78 °C] gave even worse results. In part, the latter fact can be rationalized based on decomposition reactions of **6** in etherical solvents observed in the absence of Grignard reagents (i.e., t_½_ ca. 5 min in Et_2_O at ambient temperature). Since nucleophilic substitution reactions at boron are known to proceed with higher selectivity for four-coordinate vs. three-coordinate boron compounds, borolane **6** was reacted with representative *N*-heterocyclic carbenes **7** as strong σ-donor ligands to form the borane adducts **8** [29].

The reactions of **6** with **7** afforded the adducts **8** as colorless, moisture-sensitive solids in analytically pure form. The ^11^B-NMR signals show a strong high field shift expected for a change from a three-coordinate boron atom in **6** [δ(^11^B) = 62.9 ppm] to four-fold coordination at boron [δ(^11^B) = 4.6 ppm (**8a**), 5.5 ppm (**8b**)]. X-ray crystallographic analysis for **8a** confirmed the bond connectivity, Figure 1i. The ethylene moiety C_2_H_4_ in the borolane entity of the products **8** shows only one set of protons appearing as a singlet, although the five-membered ring in the molecular structure of **8a** displays half-chair conformation, and diastereotopic protons would be expected in the ^1^H-NMR spectrum. This is remarkable since the singlet of the ethylene moietyC_2_H_4_is split up to two sets of diastereotopic protons in the final products **9** upon introduction of the alkynyl moiety (*vide infra*). We assumed a dynamic process leading to equivalent proton signals on the NMR-scale. A proposed mechanism is the dissociation of chloride in adducts **8** to give three-coordinate borenium cations of type NHC−B(S_2_C_2_H_4_)^+^ in low concentration, in which the ethylene protons can rapidly be equilibrated. [30,31] Variable temperature ^1^H-NMR experiments (CDCl_3_, −50 °C); however, did not lead to a splitting of the singlet. The ^11^B-NMR did not reveal any indication of borenium cations, which commonly give broad signals in the low field region. In part, our hypothesis is rationalized based on the mentioned singlet splitting upon substitution of chloride for the alkynyl entity, the latter of which is a poor leaving group. Reactions of NHC-adducts **8** were performed with ethynyl magnesium bromide (H−C≡C−MgBr) as a representative alkyne nucleophile. The salt elimination reactions proceeded with high selectivity and afforded the ethynyl functionalized 1,3,2-dithiaborolanes **9** as colorless solids in good yield. ^11^B-NMR spectroscopy indicated a significant high field shift upon replacement of chlorine for the ethynyl entity (i.e., 4.6 ppm (**8a**) → −11.1 ppm (**9a**), and 5.5 ppm (**8b**) → −11.5 ppm (**9b**)). The presence of ethynyl entities in the products **9** is also evident from the IR spectra (IR = infrared), in which diagnostic bands for the γ(C−H) (3220–3310 cm^−1^) and γ(C≡CH) (~2044 cm^−1^) appear and indicated the introduction of a terminal alkyne moiety. X-ray crystallographic analysis for **9a** and **9b** gave further insight into their structures, Figure 1ii,iii.

Upon replacement of chlorine in **8a** by the alkyne entity in **9a**, the bond lengths of the bound substituents increased, as indicated by the trend of the distances B1−C1 [1.6298(18) Å (**8a**) → 1.6427(15) Å (**9a**)] and the averaged distances B1−S [1.9108(14) Å (**8a**) → 1.9391(12) Å (**9a**)], which was in line with the reduced electron withdrawing effect of the alkynyl vs. chlorine substituent. The molecular structure of **9a** in the solid state revealed an interesting interaction of hydrogen atom H4 of the isopropyl group with the sulfur atom S1 of the dithioborolane cycle. The close contact S1∙∙∙H4 of 2.4620(15) Å and the orientation of the bond C4−H4 perpendicular to S1 in the dithioborolane entity indicated a weak interaction, which is at least due to van der Waals forces but may also result from weak negative hyperconjugation of a p-orbital-based lone pair at sulfur: p(S4) → σ*(C4−H4). This well ordered contact is in accordance with the strong inequality of the B−S bond lengths in **9a** (∆d(B−S)~0.06 Å) compared to **9b** (∆d(B−S)~0.005 Å), which suggests a more distorted dithioborolane cycle due to additional intramolecular interactions. Although unambiguously observed in the solid state, this bond contact is absent in solution since the NMR-spectra of compound **9a** display the expected *C_s_*-symmetry. An inspection of the bond lengths C1−B1 retrieves a slightly longer distance in **9a** (1.6427(15) Å) than in **9b** (1.6335(19) Å). This trend is expected with the higher nucleophilicity and donating ability of the *N*-heterocyclic carbene **7a** [32], resulting to give more expanded orbitals and, thus, longer bond distances at the carbene carbon atom.

## 3. Discussion

In an attempt to expand the scope of alkynyl borate thioesters [R^1^−C≡C−B(S_2_X)] from currently known compounds **4** with an aromatic backbone, we applied established routes to produce the novel alkynyl borate thioesters **5** with a saturated, aliphatic backbone. However, reactions starting from chloro dithioborolane **6** proved to be ineffective and gave inseparable mixtures of various products, as indicated by ^11^B-NMR spectroscopy of crude products. In contrast, adducts of **6** with *N*-heterocyclic carbenes (**8**) selectively react with ethynyl magnesium bromide (H−C≡C−MgBr) to afford alkynyl substituted adducts of dithioborolanes (**9**) with the aliphatic backbone in good yield. The products **9** appeared remarkably stable in air and even tolerated the work-up at aqueous conditions, which is an important aspect for future facile applications. In particular, the stability of the B−S bonds towards hydrolytic cleavage is noteworthy. Future investigation of compounds **9** will focus on the methods to further functionalize the B−S bonds with moieties of interest related to applications in materials science.

## 4. Materials and Methods

### 4.1. General Remarks

All procedures involving air or moisture sensitive compounds were performed under dry argon atmosphere using Schlenk techniques or in a glove box (MBraun 200B). Solvents were purified and dried using a solvent purification system (Braun) and stored over molecular sieve (3–4 Å). All commercially available compounds [TCI (Germany), abcr-GmbH (Karlsruhe, Germany), deutero-GmbH (Kastellaun, Germany), Sigma Aldrich (Germany)] were used without further purification. The compounds *cyclo*-ClBS_2_C_2_H_4_ (**6**) [28], IiPr (**7a**) [33], and IMes (**7b**) [34] were prepared according to literature methods.

### 4.2. Analytical Methods

NMR spectra were recorded on Bruker Avance II-300, Avance III-HD, Avance III-400, and AVII-600. The chemical shifts (δ) are reported in parts per million (ppm). The residual solvent peak (CHCl_3_, δ = 7.26 ppm) was used for referencing of the ^1^H spectra. The ^13^C spectra were internally calibrated by using the ^13^C resonances of the solvent peaks (CDCl_3_, δ = 77.16 ppm). For ^11^B NMR an external calibration with BF_3_·Et_2_O was used. Coupling constants are stated in Hertz (Hz), multiplicities are defined as br (broad), s (singlet), d (doublet), t (triplet), q (quartet), sept (septett), dd (doublet of doublets), dsept (doublet of septets), tt (triplet of triplets), tq (triplet of quartets), ddd (doublet of doublet of doublets), qdq (quartet of doublet of quartets), or m (multiplet). 2D NMR experiments (H,C-HSQC, H,C-HMBC) or gated-^13^C NMR experiments were used to aid the assignment of the signals.

Elemental analyses were accomplished by combustion and gas chromatographic analysis using a VarioMICRO Tube and thermal conductivity detection. Values are reported in %. GC-MS (gas chromatography coupled to mass spectroscopy) were recorded on a Finnigan MAT 8400-MSS I, Finnigan MAT 4515 using the EI (electron ionization) method. Values are reported as observed *m*/*z* ratio.Details for the X-ray crystallographic analysis of compounds **8a**, **9a**, and **9b** are summarized in the section “Appendix A”.

### 4.3. Synthetic Procedures

#### 4.3.1. Compound **8a**

I*i*Pr (**7a**, 400 mg, 2.63 mmol) in toluene (50 mL) was added dropwise to a solution of chloro dithioborolane (**6**, 450 mg, 3.28 mmol) in toluene (25 mL) at 0 °C. After stirring for 10 min at room temperature, the reaction mixture was dried in vacuo. The rose solid was dissolved in THF (tetrahydrofuran, 5 mL) and precipitated with pentane (20 mL). After filtration and drying in vacuo the product was obtained as a light rose solid (65% yield). Crystals suitable for X-ray crystallography were obtained by slow solvent diffusion of hexane into a solution of the product in toluene.

^1^H NMR (500.3 MHz, CDCl_3_): δ (ppm) = 7.09 (s, 2 H, NC*H*), 5.79 [sept, 2 H, C*H*(CH_3_)_2_, ^3^*J*_HH_ = 6.7 Hz], 3.12 (s, 4 H, C*H*_2_), 1.48 (d, 12 H, C*H*_3_, ^3^*J*_HH_ = 6.7 Hz).

^13^C{^1^H} NMR (125.8 MHz, CDCl_3_): δ (ppm) = 156.3 (m, N*C*N), 117.4 (s, N*C*H), 51.4 [s, *C*(CH_3_)_2_], 37.5 (s, *C*H_2_), 23.7 (s, *C*H_3_).

^11^B{^1^H} NMR (160.5 MHz, CDCl_3_): *δ* (ppm) = 4.6 (s, ω_½_ = 45 Hz).

Elemental Analysis: Calcd. for C_11_H_20_BClN_2_S_2_: C 45.45, H 6.94, N 9.64, S 22.06. Found: C 45.58, H 6.97, N 9.42, S 22.15.

#### 4.3.2. Compound **8b**

IMes (**7b**, 1.61 g, 5.28 mmol) in toluene (15 mL) was added to a solution of chloro dithioborolane (**6**, 730 mg, 5.28 mmol) in toluene (10 mL) at 0 °C. A bright yellowish solid was precipitated. The reaction mixture was stirred for 30 min at room temperature. After filtration, washing with toluene (30 mL) and subsequent drying in vacuo, the product was obtained as a white solid (93% yield). If required, further purification can be performed by slow diffusion of pentane into a solution of the crude product in CHCl_3_.

^1^H NMR (600.1 MHz, CDCl_3_): δ (ppm) = 7.10 (s, 2 H, NC*H*), 7.01 (s, 4 H, Mes-C*H*), 2.77 (s, 4 H, C*H*_2_), 2.36 (s, 6 H, Mes-*para*-C*H*_3_), 2.21 (s, 12 H, Mes-*ortho*-C*H*_3_).

^13^C{^1^H} NMR (150.9 MHz, CDCl_3_): δ (ppm) = 158.4 (m, N*C*N), 140.7 (s, Mes-*ipso*-C), 136.1 (s, Mes-*para*-*C*), 133.5 (s, Mes-*ortho*-*C*), 129.4 (s, Mes-*meta*-*C*H), 123.3 (s, N*C*H), 37.3 (s, *C*H_2_), 21.4 (s, Mes-*para*-*C*H_3_), 18.5 (s, Mes-*ortho*-*C*H_3_).

^11^B{^1^H} NMR (128.5 MHz, CDCl_3_): δ (ppm) = 5.5 (s, ω_½_ = 80 Hz).

Elemental Analysis: Calcd. for C_23_H_28_BClN_2_S_2_: C 62.38, H 6.37, N 6.33, S 14.48. Found: C 62.04, H 6.16, N 5.89, S 14.25.

#### 4.3.3. Compound **9a**

Ethynyl magnesium bromide (H−C≡C−MgBr, 1.52 mL, 0.5 M in THF, 0.76 mmol) was added dropwise to a solution of compound **8a** (200 mg, 0.69 mmol) in THF (20 mL) at 0 °C. After stirring for 15 min at 0 °C and 1 h at room temperature the reaction mixture was dried in vacuo. The remaining solids were dissolved in DCM (20 mL) and washed with distilled water (3 × 20 mL). The organic phase was dried over MgSO_4_, filtrated, and dried to yield the product as a light beige solid (75%). Crystals suitable for X-ray crystallography were obtained by storing a solution of the product in toluene at −30 °C.

^1^H NMR (500.3 MHz, CDCl_3_): δ (ppm) = 7.05 (s, 2 H, NC*H*), 6.07 [sept, 2 H, C*H*(CH_3_)_2_, ^3^*J*_HH_ = 7.0 Hz], 3.22–2.94 (m, 4 H, C*H*_2_), 2.58 (s, 1H, CC*H*), 1.46 (d, 12 H, C*H*_3_, ^3^*J*_HH_ = 7.0 Hz).

^13^C{gated-^1^H} NMR (125.8 MHz, CDCl_3_): δ (ppm) = 158.1 (br q, N*C*N, ^1^*J*_BC_ = 62.2 Hz, ^1^*J*_BC_ = 73.8 Hz), 117.3 (ddd, N*C*H, ^1^*J*_CH_ = 194.7 Hz, ^2^*J*_CH_ = 11.2 Hz, ^3^*J*_CH_ = 4.7 Hz), 99.2 (m, *C*CH), 85.9 (br d, C*C*H, ^1^*J*_CH_ = 238.7 Hz), 50.4 (dsept, *C*(CH_3_)_2_, ^1^*J*_CH_ = 144.6 Hz, ^2^*J*_CH_ = 4.2 Hz), 38.0 (tt, *C*H_2_, ^1^*J*_CH_ = 140.3 Hz, ^2^*J*_CH_ = 2.1 Hz), 23.7 (qdq, *C*H_3_, ^1^*J*_CH_ = 127.9 Hz, ^2^*J*_CH_ = 4.5 Hz, ^3^*J*_CH_ = 4.5 Hz).

^11^B{^1^H} NMR (160.5 MHz, CDCl_3_): δ (ppm) = −11.1 (s, ω_½_ = 19 Hz).

Elemental Analysis: Calcd. for C_13_H_21_BN_2_S_2_: C 55.71, H 7.55, N 10.00, S 22.88. Found: C 56.03, H 7.68, N 10.14, S 22.74.

MS (EI): *m/z* = 280.12 [M]^+^, 128.0 [M − I*i*Pr]^+^.

IR (ATR, cm^−1^): diagnostic band $\tilde{v}$ = 3306 [w, γ(CC-H)].

#### 4.3.4. Compound **9b**

Ethynyl magnesium bromide (H−C≡C−MgBr, 6.4 mL, 0.5 M in THF, 3.2 mmol) was added dropwise to a solution of compound **8b** (1.28 g, 2.9 mmol) in THF (80 mL) at 0 °C. After stirring for 30 min at 0 °C and 1 h at room temperature the reaction mixture was dried in vacuo. The remaining solids were dissolved in DCM (dichloromethane, 150 mL) and washed with distilled water (3 × 150 mL). The organic phase was dried over MgSO_4_, filtrated, and dried to yield the product as a beige solid (71%). Crystals suitable for X-ray crystallography were obtained by gas phase diffusion of pentane into a solution of the product in CDCl_3_.

^1^H NMR (600.1 MHz, CDCl_3_): δ (ppm) = 7.00 (s, 2 H, NC*H*), 6.96 (s, 4 H, Mes-C*H*), 2.77–2.37 (m, 4 H, C*H*_2_), 2.33 (s, 6 H, Mes-*para*-C*H*_3_), 2.21 (s, 12 H, Mes-*ortho*-C*H*_3_), 2.04 (s, 1 H, CC*H*).

^13^C{gated-^1^H} NMR (150.9 MHz, CDCl_3_): δ (ppm) = 161.8 (br q, N*C*N, ^1^*J*_BC_ = 61.0 Hz, ^1^*J*_BC_ = 72.8 Hz), 141.1 (q, Mes-*ipso*-C, ^3^*J*_CH_ = 6.0 Hz), 139.7 (q, Mes-*para*-*C*, ^2^*J*_CH_ = 6.0 Hz), 135.5 (q, Mes-*ortho*-*C*, ^2^*J*_CH_ = 6.0 Hz), 128.7 (m, Mes-*meta*-*C*H), 124.4 (dd, N*C*H, ^1^*J*_CH_ = 198.5 Hz, ^2^*J*_CH_ = 11.4 Hz), 97.4 (m, *C*CH), 85.6 (d, C*C*H, ^1^*J*_CH_ = 234.8 Hz), 37.3 (tt, *C*H_2_, ^1^*J*_CH_ = 140.1 Hz, ^2^*J*_CH_ = 2.8 Hz), 20.9 (tq, Mes-*para*-*C*H_3,_
^1^*J*_CH_ = 126.5 Hz, ^3^*J*_CH_ = 4.7 Hz), 18.2 (m, Mes-*ortho*-*C*H_3_).

^11^B{^1^H} NMR (160.5 MHz, CDCl_3_): δ (ppm) = −11.5 (s, ω_½_ = 30 Hz).

Elemental Analysis: Calcd. for C_25_H_29_BN_2_S_2_: C 69.43, H 6.76, N 6.48, S 14.83. Found: C 68.98, H 6.77, N 6.39, S 14.65.

MS (EI): *m/z* = 432.19 [M]^+^, 128.0 [M − IMes]^+^.

IR (ATR, cm^−1^): diagnostic bands $\tilde{v}$ = 3225 [w, γ(CC-H)], 2044 [w, γ(C≡C)].

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
