# Peer review of "N-Heterocyclic Carbene Adducts of Alkynyl Functionalized 1,3,2-Dithioborolanes"

_molecules, 2019, doi:10.3390/molecules24091690_

Round 1
Reviewer 1 Report
The communication by Frank et al. is devoted to synthesis and characterization of two NHC complexes of alkynyl functionalized 1,3,2-dithioborolanes. In general, alkynylboranes are versatile reagents therefore development of novel derivatives is desirable. In their compounds, the authors decided to replace well-known 1,3,2-dioxaborolane ring with its sulfur analogue. This offers the possibility for further functionalization at the boron atom due to a higher reactivity of boron-sulfur bond compared to the boron-oxygen one. The manuscript is concise and well written. However, some points should be clarified if the contribution could be accepted for publication.
(1) Lines 60-61: “Similar results were obtained when 6 was reacted with alkynyl organometallic reagents of more electropositive metals including R−C≡C−Li or R−C≡C−MgX.” This sentence is unclear as the R groups should be specified. Also, the type of solvent used for reactions with R−C≡C−Li should be given; it seems that organolithium reagents were added in the absence of ethereal solvents but this should be clearly stated (especially in the context of the next sentence).
(2) Lines 61-63: “In particular, in the case of Grignard reagents compound 6 was found to rapidly decompose in the presence of etherical solvents, i.e. Et2O or THF.” This sentence seems ambiguous as it is unclear whether decomposition is effected with a Grignard nucleophile (which is of course prepared in an ethereal solvent) or (maybe) with an ethereal solvent itself.
(3) Lines 95-97: “The close contact S1∙∙∙H4 of 2.4620(15) Å and the orientation of the bond C4−H4 perpendicular to S1 in the dithioborolane entity indicate a weak interaction presumably caused by negative hyperconjugation of a p-orbital based lone pair at sulfur: p(S4) → σ*(C4−H4).” I would argue about the presented interpretation. The close contact S1∙∙∙H4 can be more simply described as a weak C-H…S hydrogen bond (mostly stabilized by van der Waals forces) without involving rather specific orbital interactions.
(4) Based on the simple interpretation of the molecular structures of complexes 8a-b, the protons within each of two methylene groups of dithioborolane ring are diastereotopic which should result in two multiplets (spin system AA’BB’). However, 1H NMR spectra of 8a-b show a singlet for those CH2 protons indicating rapid inversion at the boron atom. Some discussion of a plausible mechanism of such a dynamic process is needed. In contrast, 1H NMR spectra of complexes 9a-b show two multiplets for protons of methylene groups of dithioborolane ring. This is in line with the observed chemical inertness of 9a-b as they are stable towards air and aqueous work-up. (I am wondering what is the behavior of 8a-b under such conditions?).
(5) In the supporting information, description of synthetic details and procedures (Pg 3-6) should be removed as exactly the same text is provided in the main manuscript (chapter 4. Materials and Methods). In Figures 11-12 the formulae of 9a-b lack the symbol of boron atom.
Author Response
See attached document.

Reviewer 2 Report
This paper by Frank et al describes a nice access to the interesting alkynyl thioboronates. Although use a NHC ligand appears to be necessary this work appears to pave the way for further investigation on the alkynyl thioboronates, which have not been well-understood yet. Therefore, I recommend a publication in Molecules.
Author Response
No changes requested.
Reviewer 3 Report
This manuscripts deals with the synthesis of three N-heterocyclic carbene adducts of alkynyl boronate thioesters, starting form cyclo-Cl-B(S2C2H4) and two carbenes in good yields. The synthesized compounds are analogous to the oxygen derivatives already reported and used in medicinal and material chemistry. Even short paper, the new compounds described are appropriately characterized including their X-ray molecular structure and techniques.
The abstract should be improved since is mostly devoted to introductory background and lack of conclusions.
The conclusion section is missing. I suggest only one section of results and discussion in order to include the section of conclusions.
The supplementary material should be shortened because of parts of the experimental section of the submission are duplicated. References are nested, they should be listed one by one.
Author Response
See attached document.

Round 2
Reviewer 1 Report
The authors fully addressed my comments and therefore I would recommend the acceptance of the revised manuscript for publication.